# Effectiveness of Gabapentin as a Benzodiazepine-Sparing Agent in Alcohol Withdrawal Syndrome

**DOI:** 10.3390/medicina60061004

**Published:** 2024-06-19

**Authors:** Hamza Alzghoul, Mohammed I. Al-Said, Omar Obeidat, Hashim Al-Ani, Mohammad Tarawneh, Robyn Meadows, Houssein Youness, Raju Reddy, Mohammad Al-Jafari, Bashar N. Alzghoul, Akram Khan

**Affiliations:** 1Graduate Medical Education, University of Central Florida College of Medicine, Orlando, FL 32816, USA; hamza.alzghoul@hcahealthcare.com; 2Department of Pharmacy, HCA Florida North Florida Hospital, 6500 W Newberry Rd, Gainesville, FL 32605, USA; mohammed.alsaid@hcahealthcare.com; 3Internal Medicine Residency Program, HCA Florida North Florida Hospital, 6500 W Newberry Rd, Gainesville, FL 32605, USA; omar.obeidat@hcahealthcare.com (O.O.); hashim.alani@hcahealthcare.com (H.A.-A.); mohdtarawneh1@gmail.com (M.T.); 4Graduate Medical Education, HCA Healthcare, Brentwood, TN 37027, USA; robyn.meadows@hcahealthcare.com; 5Section of Pulmonary, Critical Care and Sleep Medicine, The Oklahoma City VA Health Care System and The University of Oklahoma Health Sciences Center, Oklahoma City, OK 73104, USA; houssein-youness@ouhsc.edu; 6Department of Internal Medicine, Oregon Health and Science University, Portland, OR 97239, USA; raju.reddy@austin.utexas.edu; 7Faculty of Medicine, Mutah University, Al-Karak 61710, Jordan; mhmmdaljafari@gmail.com; 8Division of Pulmonary, Critical Care and Sleep Medicine, University of Florida, Gainesville, FL 32608, USA; bashar.alzghoul@medicine.ufl.edu

**Keywords:** alcohol withdrawal syndrome, gabapentin, benzodiazepines

## Abstract

*Background and Objectives:* Gabapentin has shown promise as a potential agent for the treatment of alcohol withdrawal syndrome. We aimed to evaluate the effectiveness of gabapentin as a benzodiazepine-sparing agent in patients undergoing alcohol withdrawal treatment in all the hospitals of a large tertiary healthcare system. *Materials and Methods:* Medical records of patients admitted to the hospital for alcohol withdrawal management between 1 January 2020 and 31 August 2022 were reviewed. Patients were divided into two cohorts: benzodiazepine-only treatment who received benzodiazepines as the primary pharmacotherapy and gabapentin adjunctive treatment who received gabapentin in addition to benzodiazepines. The outcomes assessed included the total benzodiazepine dosage administered during the treatment and the length of hospital stay. The statistical models were calibrated to account for various factors. *Results:* A total of 4364 patients were included in the final analysis. Among these, 79 patients (1.8%) received gabapentin in addition to benzodiazepines, and 4285 patients (98.2%) received benzodiazepines only. Patients administered gabapentin required significantly lower average cumulative benzodiazepine dosages, approximately 17.9% less, compared to those not receiving gabapentin (median 2 mg vs. 4 mg of lorazepam equivalent dose (*p* < 0.01)). However, there were no significant differences in outcomes between the two groups. *Conclusions:* Our findings demonstrate that using gabapentin with benzodiazepine was associated with a reduction in the cumulative benzodiazepine dosage for alcohol withdrawal. Considering gabapentin as an adjunctive therapy holds promise for patients with comorbidities who could benefit from reducing benzodiazepine dose. This strategy warrants further investigation.

## 1. Introduction

Alcohol consumption has significant implications for healthcare in terms of cost and individual well-being. In 2018, approximately 60–70% of adults in the United States reported consuming alcohol, with 1 in 12 adults engaging in heavy drinking [1]. Alcohol withdrawal (AW) requires pharmacologic therapy in nearly half a million cases annually in the United States [2]. Symptoms of AW can range from mild tremors and anxiety to severe complications such as hallucinations, seizures, delirium tremens, and life-threatening metabolic issues [3].

In AW, the sudden cessation of prolonged alcohol intake may cause a brisk hyperactivation of N-methyl-D Aspartate (NMDA) receptors, causing devastating autonomic symptoms [2,3]. Effective treatment regimens for AW are crucial for symptom control and as a starting point for long-term treatment of alcohol use disorder [4]. Benzodiazepines such as diazepam are considered the preferred treatment for AW symptoms due to their mechanism of action [5]. However, the potential side effects of these agents, such as respiratory depression and addiction potential, may limit their use in large doses [2].

Gabapentin, an analog of gamma-aminobutyric acid (GABA) is used as an antiepileptic agent to treat postherpetic neuralgia and restless leg syndrome [3,6]. This agent was shown to have the potential to relieve the debilitating symptoms of seizures, delirium, and increased autonomic tone in AW. It has been suggested that a combined use of benzodiazepines and gabapentin may lead to a decrease in the required dose of benzodiazepines needed to treat AW symptoms [5]. The mechanism of action of gabapentin is thought to be mediated by increasing GABA synthesis, decreasing glutamate transmission, and blocking the neuronal calcium channels [6]. Moreover, its secondary effects of excitatory neurotransmission blockade and modifying glutamic acid decarboxylase enzyme to decrease glutamate synthesis render it a valuable and promising agent [7]. Interestingly, at relatively higher doses, gabapentin was associated with even better results than benzodiazepines in terms of quality of sleep, controlling anxiety, and returning to work intervals [7].

In the past few years, gabapentin has drawn increasing attention. A few trials looking at either alcohol dependence or AW have shown promising results for the use of gabapentin as a single agent, but most of these studies were in the outpatient setting [8]. The data regarding the use of gabapentin for AW in the inpatient setting is, however, limited [2,9,10,11,12,13]. For example, Morrison et al. concluded that using gabapentin in AW in the inpatient setting was associated with decreased in-hospital length of stay, a 30% reduction in lorazepam use, and decreased duration of delirium [6].

In this study, we aimed to evaluate the effectiveness of gabapentin as an adjunctive treatment to benzodiazepines for managing alcohol withdrawal in hospitalized patients, including those with moderate and severe presentations. Specifically, we investigated whether the addition of gabapentin can reduce the cumulative benzodiazepine dosage.

## 2. Materials and Methods

This is a retrospective study performed using data from a large tertiary healthcare system, the Hospital Corporation of America (HCA). In 2020, gabapentin was added to some of the HCA hospital’s inpatient AW management protocol, which is guided by the Clinical Institute Withdrawal Assessment of Alcohol scale, Revised (CIWA-AR) (Appendix A) [14]. To study the effectiveness of this new protocol, we conducted a retrospective cohort-matched study from 1 January 2020 to 31 August 2022, including patients who were admitted to any HCA hospital with the diagnosis of AW syndrome being among the top 3 diagnoses, identified using ICD-10 codes. Patients were divided into two groups based on their treatment for AW: (1) a gabapentin and benzodiazepine treatment group, who received gabapentin as an adjunct to benzodiazepines, and (2) a benzodiazepine-only group.

The gabapentin and benzodiazepine treatment group included patients who were initiated on benzodiazepines for AW according to the CIWA protocol along with gabapentin. To be included in this group, patients had to receive at least 900 mg of gabapentin on day 1 or an average of 900 mg or more/day over their stay. Lower dosages were excluded in accordance with the existing literature, which has used a dose limit higher than 900 mg daily on average for the entire course of treatment [2,9,10,11,12,13]. The benzodiazepines-only group included patients who received benzodiazepines but did not receive any dosage of gabapentin for AW management or any other reasons.

We included patients who were 18 years old or older and admitted to one of the HCA hospitals for more than one day with AW by ICD-10 codes and received gabapentin or benzodiazepine during their hospital stay. We excluded patients who were <18 years old, prescribed gabapentin, pregabalin, or benzodiazepines in the outpatient setting, or had missing or extreme (more than 150 days) length of hospital stay; missing data included missing dosages of the involved medications and missing patient characteristics (Figure 1). Our study was limited to gabapentin- and benzodiazepine-naive patients.

The following patient data were obtained: age, sex, race (white vs. non-white), BMI, length of hospital stay, length of ICU stay, comorbidities (Elixhauser comorbidity index) [15,16], and the cumulative benzodiazepine dose (lorazepam equivalent dose) (Appendix A) [17]. The primary outcome of this study was the cumulative benzodiazepine dose described as lorazepam equivalent. The secondary outcomes included changes in CIWA score between admission and at 48 h and day 5, in-hospital mortality, length of stay, and length of ICU admission. Data were analyzed and compared between the two groups. This research activity was determined to be exempt or excluded from Institutional Review Board (IRB) oversight in accordance with current regulations and institutional policy. Our internal reference number for this determination is #2022-958. Patient data were de-identified and analyzed on a secured server. The STROBE (Strengthening the Reporting of Observational Studies in Epidemiology) guidelines were followed in this study [18].

### 2.1. Statistical Analysis

A dedicated statistician conducted data extraction and analyses. Descriptive statistics, including means, medians, standard deviations, percentages, and frequencies, were utilized to describe the baseline characteristics of the patient population. Inferential statistics were used to test the differences between the two medication groups for various outcomes. Mixed-model ANOVA tests were employed to test the difference between the two medication groups concerning the CIWA score, with day included as a within-subjects variable and medication group included as a between-subjects variable. In addition, a general linear model was utilized to test the difference between the two medication groups concerning the cumulative benzodiazepine dose, length of hospital stay, and length of ICU stay. Given that the data are non-normally distributed, the median serves as a more suitable comparison parameter. It is important to note that the cumulative benzodiazepine dose was log-transformed to fulfill normality assumptions for the general linear model. Whenever an outcome variable is log-transformed, you should exponentiate the parameter estimates [e] to find the percent difference, which equals ((e × 100) − 100). The models were calibrated to account for various factors, including the duration of hospitalization, age, gender, ethnicity, metric-BMI, presence of chronic kidney disease (CKD), liver failure, generalized anxiety disorder, and a modified Elixhauser comorbidity score. In the modified Elixhauser score, we excluded alcohol abuse, CKD, and liver disease, which are typically included in the index, and added them as separate control variables. Associations were considered significant if the *p*-value was less than 0.05. Data were analyzed using SAS (version 9.4, Cary, NC, USA).

### 2.2. Ethics

Informed consent was waived by the IRB due to the retrospective nature of this study. This research study, with IRB Number 2022-958, was conducted following ethical principles and guidelines. The research activity was reviewed by the Institutional Review Board (IRB) in accordance with current regulations and institutional policy. Based on the review, this research was deemed exempt or excluded from IRB oversight, as it posed minimal risk to human subjects and involved the analysis of existing anonymized data. The exemption was granted with the internal reference number 2022-958, signifying this study’s compliance with ethical standards and guidelines. The researchers ensured the confidentiality and privacy of the participants’ information throughout this study and adhered to the principles outlined in the Declaration of Helsinki.

## 3. Results

### 3.1. Data Excluded from This Study

A total of 9535 patients with AW were identified between 1 January 2020 and 31 August 2022. Of these, 8 patients were excluded due to a lack of demographic information, and 4838 patients were excluded due to prior use of gabapentin, pregabalin, or benzodiazepines, leaving 4689 benzodiazepine- and gabapentin-naive patients as the initial sample. Subsequently, 207 patients were excluded due to missing data needed to calculate equivalency, and 76 patients were excluded due to low-dose gabapentin use (<900 mg/day). An additional 42 patients were excluded due to extreme length of stay (>150 days) or extreme doses of benzodiazepine (>70 mg of lorazepam equivalent dose over the entire hospital stay). Figure 1 illustrates the flowchart of patient exclusions.

### 3.2. Population Characteristics

Table 1 presents the baseline characteristics of the patient cohort based on their medication group. This study included 4364 patients, out of which 79 patients (1.8%) received gabapentin, and 4285 patients (98.2%) received benzodiazepines. Patient sex, race, and comorbidities are detailed in (Table 1). There were no statistically significant differences between the two groups in terms of age, sex, race, and comorbidities.

Out of the 4364 patients, 435 (10%) were admitted to the ICU, with 425 (9.9%) in the benzodiazepine group and 10 (12.7%) in the gabapentin group. The average length of stay in the hospital was 4.4 days (+/−3.9), and the average length of stay in the ICU was 2.3 days (+/−2.4). The mortality rate was 0.23%, with no deaths reported in the gabapentin group and ten deaths reported in the benzodiazepine group.

### 3.3. Benzodiazepine Dose and Medication Group

The cumulative benzodiazepine dosage calculated as a lorazepam equivalent dose was determined for each patient, with the entire population demonstrating an average dose of 7.5 (+/−9.5) mg [16]. Patients who received gabapentin required a significantly lower dose of benzodiazepines in lorazepam equivalents. When controlling for other variables, those administered gabapentin received an average of 17.9% less benzodiazepines (median = 2 mg) compared to those who were not administered gabapentin (median 4 mg) (*p* < 0.01). To explain the 17.9%, it is crucial to understand that the cumulative benzodiazepine dose was log-transformed to fulfill normality assumptions for the general linear model; calculation of the percentage difference in the setting of log transformation is explained in the Section 2. Although the mean benzodiazepine dosage was higher in the gabapentin group (8.6 mg vs. 7.4 mg), the median dosage was 50% lower (2 mg vs. 4 mg, *p* < 0.01). Additionally, multiple factors, including length of hospital stay, sex, race, the presence of liver disease, the presence of generalized anxiety disorder, and Elixhauser comorbidity score, had a significant impact on the cumulative dosage of benzodiazepines (Table 2).

### 3.4. CIWA Score Change and Medication Group

In total, 1815 patients had CIWA scores available at day 0 and day 2, and 340 patients had available CIWA scores at day 0, day 2, and day 5. Regarding the severity of AW, the mean CIWA score on day 0 was 9.0 (+/−12.9) for the entire population. The gabapentin group had a higher mean score of 15.4 (+/−21.2), while the benzodiazepine group had a lower mean score of 8.9 (+/−12.7). On day 2, the mean CIWA score was 11 (+/−12.2) for the entire population, with a lower mean score of 4.9 (+/−7) for the gabapentin group and a higher mean score of 11.1 (+/−12.2) for the benzodiazepine group. On day 5, the mean CIWA score was 5.6 (+/−10.4) for the entire population, with a slightly higher mean score of 6.2 (+/−7.7) for the gabapentin group and a similar mean score of 5.6 (+/−10.4) for the benzodiazepine group.

The results indicated that there was no significant difference in the CIWA score between the medication groups (*p* = 0.98), and there was also no significant difference in the CIWA score between day 0 and day 2 (*p* = 0.19). Additionally, the interaction between the two factors (day and medication group) was not significant (*p* = 0.45).

Subsequently, an additional model was implemented that included day 5 in the analysis. The findings revealed that there was no significant difference in the CIWA score between the medication groups (*p* = 0.93). Furthermore, no significant difference in the CIWA score was observed between day 0, day 2, and day 5 (*p* = 0.18) (Figure 2). The interaction between the medication group and time factor was also not statistically significant (*p* = 0.4).

### 3.5. Length of Stay, ICU Length of Stay, and Medication Group

The impact of medication group and other variables on the length of stay and ICU stay were analyzed using general linear models. The results indicated that there was no significant difference in the average length of stay between medication groups (*p* = 0.33) when controlling for other variables. However, several other factors, such as BMI (*p* = 0.004), the presence of generalized anxiety disorder (*p* < 0.001), and Elixhauser comorbidity score (*p* < 0.001), were found to significantly impact the length of stay (see Appendix A).

Similarly, the average ICU length of stay did not significantly differ between medication groups (*p* = 0.47) when controlling for other variables. Other factors, such as age (*p* = 0.04), sex (*p* = 0.04), and Elixhauser comorbidity score (*p* < 0.001), were observed to significantly impact the length of ICU stay (see Appendix A).

## 4. Discussion

We found that, after controlling for confounding variables, patients administered gabapentin in addition to benzodiazepines received a 17.9% lower dose of benzodiazepines compared to those who were not administered gabapentin. Our study is unique as it includes patients with severe alcohol withdrawal and is limited to those who were naive to gabapentin, pregabalin, and benzodiazepines.

Mild alcohol withdrawal is characterized by mild or moderate anxiety, sweating, and insomnia but no tremors (CIWA-Ar < 10). In contrast, moderate alcohol withdrawal is characterized by moderate anxiety, sweating, insomnia, and mild tremors (CIWA-Ar 10-18) [19,20]. We highlight that the gabapentin with benzodiazepine group had a higher baseline CIWA score compared to the benzodiazepine-only group, indicating more severe withdrawal symptoms. Despite this, patients in the gabapentin group received a significantly lower cumulative dose of benzodiazepines. Importantly, the CIWA scores between the gabapentin with benzodiazepine and benzodiazepine-only groups were similar despite the reduced benzodiazepine usage in those who received gabapentin as an adjunct.

The American Society of Addiction Medicine (ASAM) recommends gabapentin for treating withdrawal symptoms as follows: (I) Gabapentin is an appropriate alternative to benzodiazepines for mild to moderate alcohol withdrawal. (II) Gabapentin is a suitable choice for treating alcohol withdrawal when the physician also intends to utilize it for a patient’s ongoing treatment of alcohol use disorder. (III) Gabapentin may be used as an adjuvant to benzodiazepine therapy to help reduce alcohol withdrawal. However, before doing so, doctors should ensure that an adequate amount of benzodiazepine has been delivered. (IV) If benzodiazepines are not tolerated, gabapentin is a suitable choice for patients with mild or moderate withdrawal [19,20]. In these intolerant patient subgroups, it is essential to explore benzodiazepine-sparing agents, such as gabapentin, for managing alcohol withdrawal, particularly due to their lower addictive potential and ability to reduce alcohol cravings, thereby facilitating abstinence [7].

The results of studies that used gabapentin alongside CIWA-directed benzodiazepines have been inconsistent. Similar to our study, a retrospective study comparing 50 patients receiving gabapentin at least 1800 mg per day in the first 2 days of AW syndrome to 50 patients who received benzodiazepines discovered that the gabapentin group needed fewer benzodiazepine doses [11]. In contrast, three retrospective studies found no decrease in benzodiazepine use [2,12,13]. However, it is important to highlight that patients in the Anduluz et al. study who received a total of 2100 mg of gabapentin on day 1 and 1800 mg on day 2 were older and experienced less severe AW symptoms than the benzodiazepine group in this study; patients in the gabapentin group had an average higher CIWA-Ar score of 7.9 vs. 6.1 in the control group. On the other hand, our gabapentin group had a baseline average CIWA score of 15.3 vs. 8.9 in the control group [2]. In addition, a recent retrospective study revealed that gabapentin did not affect the number of benzodiazepine doses that were required to treat AW symptoms [12]. These differences can be attributed to differences in baseline variables across studies and to the different dosage cutoffs of medications used in studies. In addition, the comparison of patients from different time periods may have introduced possible confounders. In order to mitigate these potential biases, we only included patients who were naive to gabapentin, pregabalin, and benzodiazepines. Patients who were receiving any of these medications as outpatients were excluded. In addition, our analysis models were adjusted for factors like hospitalization duration, age, gender, ethnicity, and comorbidities. This approach helps isolate the effect of the medications from other factors. Furthermore, our study was similar to Anduluz et al., Nichols elik et al., and Vadiei et al., where gabapentin treatment was adjunctive to the CIWA score-based benzodiazepine administration [2,9,10].

Defoster et al. recently conducted a randomized controlled trial in the inpatient setting, using fixed-dose gabapentin taper and CIWA-directed benzodiazepines to treat mild/moderate AWS [21]. Their study found no significant differences between the groups; however, the results were constrained by under-enrollment and prior benzodiazepine use in both groups. Notably, this study excluded patients with severe withdrawal symptoms and ICU admissions. In contrast, our study included patients with all severities of withdrawal, including those admitted to the ICU, potentially providing insights into the efficacy of gabapentin in more severely affected populations.

In our study, the mortality rate was 0.2%, with no deaths reported in the gabapentin group and ten deaths reported in the benzodiazepine group. According to the current literature, gabapentin has lower risks of adverse effects and frequently reduces withdrawal symptoms, such as cravings, anxiety, and insomnia [7]. According to a Mayo Clinic retrospective study, 148 patients were moved from the benzodiazepine group to the gabapentin group due to clinical worsening, psychiatric advice, or to decrease cravings and facilitate abstinence after discharge [9]. While due to the retrospective nature of our study, we cannot definitely conclude whether gabapentin therapy is safe in all individuals with all severities of AW syndrome, we found no significant difference in the CIWA score between day 0, day 2, and day 5 among the intervention and control groups. It is important to note that CIWA scores were available for only 340 patients on day 5, while 1815 patients had scores recorded on day 0 and day 2. This pattern is likely due to patients improving over time; as they recover from withdrawal symptoms, the nursing staff stopped administering CIWA scores. This led to a reduction in the number of patients with CIWA scores on day 5. Therefore, it is reasonable to assume that only patients experiencing severe and prolonged withdrawal symptoms are the ones who remain symptomatic and continue to have CIWA scores recorded by the end of their withdrawal period. A double-blind, controlled trial carried out on outpatients compared various gabapentin dosage regimens with lorazepam and found that a gabapentin dose of 1200 mg was more effective than lorazepam in a fixed-dosed strategy for reducing AW symptoms [22]. However, given their inclusion criteria, the findings may not be clinically applied to patients requiring hospitalization or those experiencing severe acute AW symptoms. Additionally, this study’s relatively small sample size of 100 patients and the further randomization of these patients to one of four treatment arms limited its statistical power.

In our study, there was no significant difference in the average length of stay, including the ICU length of stay, between the two medication groups. However, factors including BMI, generalized anxiety disorder, and the comorbidity score were found to have a significant impact on the length of stay between the two groups. Aligning with our findings, the same efficacy and safety were seen in the intervention and control groups in a Mayo Clinic review of gabapentin for acute withdrawal symptoms in 77 inpatients. Nevertheless, the two groups had no statistically significant difference in the length of stay [10]. On the other hand, a recent retrospective study found that patients in the gabapentin group were more likely to be discharged earlier compared to the benzodiazepine group [11]. They concluded that taking gabapentin had been associated with a 1.4-day reduction in hospital stay compared to the control group. These varying results highlight the diverse evidence related to our research question, underscoring the need for additional large-scale, controlled, and prospective studies to better define groups that will benefit from early administration of gabapentin.

Our study has some limitations. First, this study carries the inherent limitations of retrospective observational studies. Secondly, with only 79 of the 4364 patients in our study receiving gabapentin, there is a potential risk of bias in the results. Finally, we could not assess post-discharge outcomes to determine recurrence or readmission rates. Despite these limitations, our study has several strengths and adds to the knowledge on this topic. In order to mitigate potential biases, we focused exclusively on patients who were gabapentin and benzodiazepine naive. In addition, we accounted for several confounding variables, including length of stay, age, sex, race, BMI, CKD, liver failure, generalized anxiety disorder, and the Elixhauser comorbidity index score, which could potentially influence the outcome. Furthermore, our data were collected from multiple hospitals within the HCA healthcare system, spanning multiple states and covering a diverse population compared to other retrospective studies on this topic which were conducted in a single-center setting.

## 5. Conclusions

Our findings demonstrate that using gabapentin with benzodiazepine was associated with reduced benzodiazepine dosage. However, no significant difference was observed in the length of hospital stay or the change in CIWA score on day 2 or day 5. Future research, especially randomized controlled trials, would be required to confirm these findings and identify potential patient subgroups, such as those with comorbidities who might benefit the most from adjunctive gabapentin therapy. Additionally, future studies should address our research limitations by including larger sample sizes to reduce potential bias and incorporating follow-up assessments to evaluate post-discharge outcomes. Investigating the long-term effects of gabapentin across different patient demographics and settings would also provide valuable insights into its broader applicability in managing alcohol withdrawal.

## Figures and Tables

**Figure 1 medicina-60-01004-f001:**
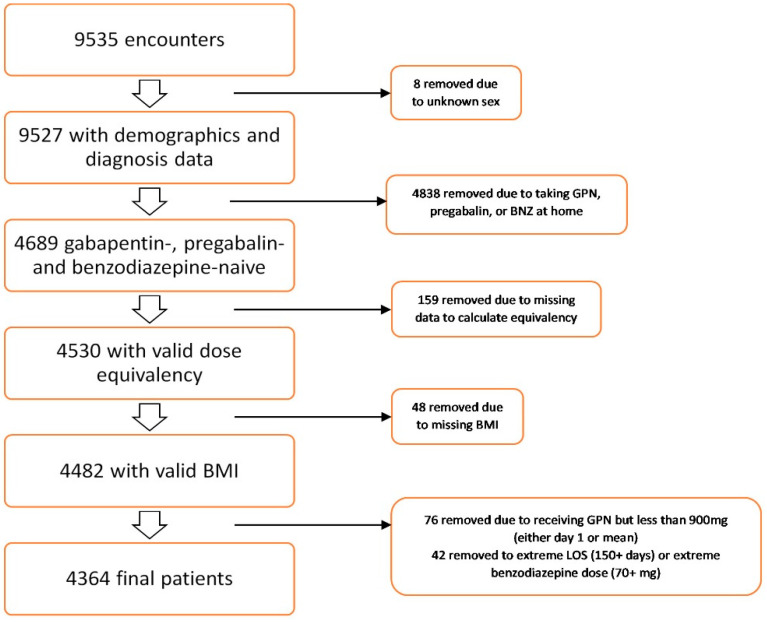
Data excluded from this study.

**Figure 2 medicina-60-01004-f002:**
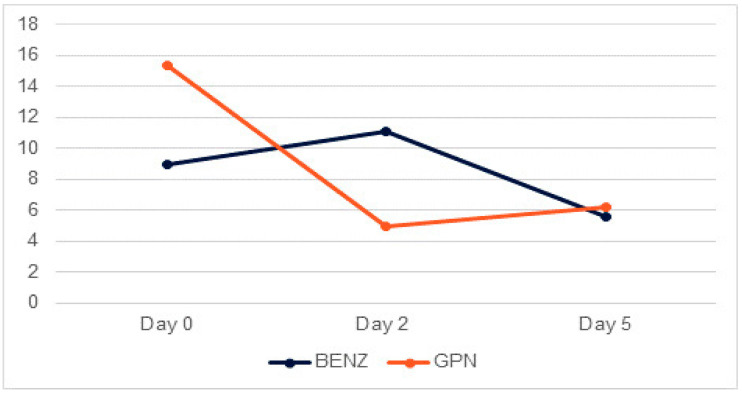
CIWA score change between day 0, day 2, and day 5 for benzodiazepine-only group (*N* = 333) compared to gabapentin plus benzodiazepine group (*N* = 7). BENZ: benzodiazepines group. GPN: gabapentin plus benzodiazepines group.

**Table 1 medicina-60-01004-t001:** Baseline characteristics of patients stratified by medication group.

Variable	Level	Total	Med Group		
			GPN	BNZ	*p*-Value
Patients [N (%)]		4364 (100.00)	79 (1.81)	4285 (98.19)	0.876
Sex [N (%)]	M	3003 (68.81)	55 (69.62)	2948 (68.80)	
	F	1361 (31.19)	24 (30.38)	1337 (31.20)	
Age [M (SD)]		46.02 (15.10)	43.99 (12.67)	46.06 (15.14)	0.155
Race [N (%)]	White	3024 (69.29)	60 (75.95)	2964 (69.17)	0.196
	Nonwhite	1340 (30.71)	19 (24.05)	1321 (30.83)	
Comorbidities [N (%)]	CKD	112 (2.57)	3 (3.80)	109 (2.54)	0.457
	Liver Disease	697 (15.97)	11 (13.92)	686 (16.01)	0.616
	GAD	1251 (28.67)	23 (29.11)	1228 (28.66)	0.929
Elixhauser * [M (SD)]		2.16 (1.40)	2.10 (1.57)	2.16 (1.39)	0.695
LOS [Med (IQR)]		3.00 (2.00, 5.00)	3.00 (2.00, 5.00)	3.00 (2.00, 5.00)	0.331
ICU LOS [Med (IQR)]		2.00 (1.00, 3.00)	2.00 (1.00, 6.00)	2.00 (1.00, 3.00)	0.570
Mortality [N (%)]		10 (0.23)	0 (0.00)	10 (0.23)	1.000
CIWA Day 0 [Med (IQR)]		3.25 (1.00, 6.67)	2.00 (0.08, 5.75)	3.25 (1.00, 6.67)	0.076
CIWA Day 2 [Med (IQR)]		1.67 (0.00, 4.60)	0.84 (0.00, 2.90)	1.67 (0.00, 4.67)	0.145
CIWA Day 5 [Med (IQR)]		1.00 (0.00, 3.00)	0.91 (0.33, 1.57)	1.00 (0.00, 3.00)	0.899
Cum. BNZ Dose [Med (IQR)]		4.00 (2.00, 9.50)	2.00 (0.00, 11.00)	4.00 (2.00, 9.40)	0.0006

* Elixhauser comorbidity index modified to exclude alcohol abuse due to patient population and to exclude renal failure (moderate and severe) and liver disease (moderate and severe); BNZ: benzodiazepine, CIWA: Clinical Institute Withdrawal Assessment of Alcohol scale, CKD: chronic kidney disease, GAD: generalized anxiety disorder, GPN: gabapentin, LOS: length of stay, ICU: intensive care unit, IQR: interquartile range, M: mean, Med: median, N: number, SD: standard deviation. For cumulative benzodiazepine dose, CIWA scores, LOS, or ICU LOS. All of these variables are our outcome variables and have *p*-values provided in the manuscript adjusted for the appropriate control variables. Including *p*-values on this table would provide an unadjusted *p*-value, which is not as informative and could be misinterpreted.

**Table 2 medicina-60-01004-t002:** General linear model for cumulative benzodiazepine dose.

Effect	Num DF	Den DF	Estimate	Standard Error	F Value	Pr > F
Medication Group	1	79	−0.1643	0.05927	7.68	0.0069
LOS *	1	4323	0.01480	0.001471	101.25	<0.0001
Age	1	4322	0.000043	0.000402	0.01	0.9143
Sex	1	4332	−0.03401	0.01225	7.70	0.0055
Race	1	4329	−0.06797	0.01234	30.32	<0.0001
BMI	1	4322	0.000301	0.000979	0.09	0.7585
CKD	1	4347	−0.03099	0.03622	0.73	0.3923
Liver Disease	1	4330	0.09863	0.01545	40.76	<0.0001
GAD	1	4336	0.03245	0.01279	6.44	0.0112
ELIX	1	4341	0.02246	0.004338	26.80	<0.0001

* Length of stay is included in this model as a covariate to control for the scaling of the outcome; LOS: length of stay, BMI: body mass index, CKD: chronic kidney disease, GAD: generalized anxiety disorder, ELIX: Elixhauser comorbidity index.

## Data Availability

Data supporting the reported results are provided in the article. Additional deidentified data are available upon request to the corresponding author.

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
