# Peer review of "Effectiveness of Gabapentin as a Benzodiazepine-Sparing Agent in Alcohol Withdrawal Syndrome"

_medicina, 2024, doi:10.3390/medicina60061004_

Round 1

Reviewer 1 Report

Comments and Suggestions for Authors

Dear authors,

Thank you for your submission. Please find my comments below.

At the end of the discussion, please mention the research hypothesis. As I can see, the hypothesis is mentioned in the first paragraph of the discussion. Please change that.

The methodology is inconsistent- patients had to receive at least 900mg of gabapentin on day 1 or an average of 900mg or more/day over their stay. 

There is a gross disparity between sample size in both groups-, 79 patients (1.8%) received gabapentin, and 4,285 patients (98.2%) received benzodiazepines only. It is very difficult to conclude convincingly with such a gross discrepancy in the sample size between either group. Although this limitation has been mentioned, still it is extremely difficult to accept the results of this study in clinical practice.

The authors conclude that patients administered gabapentin in addition to benzodiazepines received a 17.9% lower dose of benzodiazepines compared to those who were not administered gabapentin. However, the results mention that this dose reduction does not impact any clinically relevant outcomes. Therefore, the results of this study are not relevant to clinical practice.

Comments on the Quality of English Language

Overall, the use of English language is satisfactory.

Reviewer 2 Report

Comments and Suggestions for Authors

Dear authors,

Many thanks for submitting your manuscript to the journal. In general, I reviewed a well-structured manuscript investigating an interesting research topic. However, you need to address some significant concerns that could negatively impact the quality of your work.

I have now uploaded my evaluation report.

Best regards

Round 2

Reviewer 1 Report

Comments and Suggestions for Authors

Dear authors,

Thank you for editing your submission based on the comments raised. The comments were meant to improve the quality of your submission and were not intended to criticize in any way. I wish you all the best.

Comments on the Quality of English Language

Minor technical edits will be required.

Reviewer 2 Report

Comments and Suggestions for Authors

Dear authors,
Many thanks for providing the revised version of your manuscript. Many issues have been effectively addressed. However, the inherent weakness of your study, the small number of cases compared to controls remains. The final decision depends on the journal's standards and policies.

Best regards,